# The Role of the Reanalysis of Genetic Test Results in the Diagnosis of Dysmorphic Syndrome Caused by Inherited Xq24 Deletion including the *UBE2A* and *CXorf56* Genes

**DOI:** 10.3390/genes12030350

**Published:** 2021-02-27

**Authors:** Ewelina Wolańska, Agnieszka Pollak, Małgorzata Rydzanicz, Karolina Pesz, Magdalena Kłaniewska, Anna Rozensztrauch, Paweł Skiba, Piotr Stawiński, Rafał Płoski, Robert Śmigiel

**Affiliations:** 1Department of Pediatrics, Division of Propaedeutic of Pediatrics and Rare Disorders, Wroclaw Medical University, 51-618 Wroclaw, Poland; magdalena.klaniewska@student.umed.wroc.pl (M.K.); robert.smigiel@umed.wroc.pl (R.Ś.); 2Department of Medical Genetics, Warsaw Medical University, 02-106 Warsaw, Poland; poli25@wp.pl (A.P.); mrydzanicz@wum.edu.pl (M.R.); stawinski84@gmail.com (P.S.); rploski@wp.pl (R.P.); 3Department of Genetics, Wroclaw Medical University, 50-368 Wroclaw, Poland; karolina.pesz@umed.wroc.pl (K.P.); pawel.skiba@umed.wroc.pl (P.S.); 4Department of Pediatrics, Division of Neonatology, Wroclaw Medical University, 51-618 Wroclaw, Poland; anna.rozensztrauch@umed.wroc.pl

**Keywords:** *UBE2A* gene, *CXorf56* gene, contiguous gene deletion Xq24, intellectual disability, dysmorphic syndrome

## Abstract

Psychomotor delay, hypotonia, and intellectual disability, as well as heart defects, urogenital malformations, and characteristic cranio-facial dysmorphism are the main symptoms of dysmorphic syndrome associated with intergenic deletion in the Xq24 chromosome region including the *UBE2A* and *CXorf56* genes. To date, there is limited information in the literature about the symptoms and clinical course of the Xq24 deletion. Here, we present a case of Xq24 deletion including the *UBE2A* and *CXorf56* genes in a nine-year-old boy, in whom the array comparative genomic hybridization (array-CGH) and whole exome sequencing (WES) tests were performed in 2015 with normal results. The WES results were reanalyzed in 2019. Intergenic, hemizygous deletion in the Xq24 chromosome region including the *UBE2A* and *CXorf56* genes was revealed and subsequently confirmed in the array-CGH study as the deletion of 35kb in the Xq24 region. Additionally, the carriership of deletion in the mother of the child was confirmed.

## 1. Clinical Report

The eight-year-old boy was born at term, after an uneventful pregnancy, with a birth weight of 4100 g (50–90 percentile). He received 10 points on the Apgar score. The prenatal history was negative for known teratogens. On the initial physical examination, the boy presented with craniofacial dysmorphic features. A large tongue, webbed neck, deformation of the chest, abnormalities of the hands, and genitourinary system anomalies (hypospadias and cryptorchidism) were also present. Additionally, heart defects (atrial septal defect, ventricular septal defect) were found. Both the heart defects and cryptorchidism required surgical procedures. There was a significant psychomotor development delay. The boy began to walk with an aid at the age of eight and did not develop speech. Aggressive behavior and hyperactivity were observed from the age of four. He was also diagnosed with drug-resistant epilepsy at the age of four. Contrast magnetic resonance imaging (MRI) of the brain showed non-specific demyelinating or dysmyelinating lesions with a vascular component and a mild lateral ventricular dilatation (Evans Index 0.32). The thyroid hormone level was normal. Additionally, at the age of eight, joint contractures, excessive skin elasticity, and decreased muscle tone were observed. The dysmorphic features of the patient were described as follows: broad face with short and broad neck, low posterior hairline, large ears, prominent supraorbital ridges, broad eyebrows and long palpebral fissures, long lashes, and wide mouth with large tongue. Other findings included widely spaced nipples, broad great toes in the valgus position, as well as short hands and feet with small fingernails (Figure 1).

The overall clinical picture was suggestive of a genetic condition from the very beginning. Up to age five, numerous genetic tests, including karyotyping, methylation test, array comparative genomic hybridization (array-CGH), and whole exome sequencing (WES), were performed to no avail. Despite the negative results of all previous genetic tests, at the age of eight, the patient was re-evaluated by a clinical geneticist, which was followed by WES reanalysis, and another array-CGH was performed with the indication of the causative deletion on chromosome Xq24.

## 2. Genetic Results

### WES Study

The DNA of the proband and his parents was isolated from peripheral blood lymphocytes, whereas the grandparents’ DNA was isolated from buccal swabs, all according to the standard protocols. The library was prepared using the SureSelect Human All Exon v5 kit (Agilent, Santa Clara, CA, USA) and paired-end sequenced (2 × 100bp) on a HiSeq 1500 (Illumina, San Diego, CA, USA). Bioinformatic analysis of raw WES data and variants prioritization were performed as previously described [1]. Reads were aligned to the hg38 reference genome sequence. Integrative Genomics Viewer v.2.8 was used to visualize the WES results (IGV) [2]. After the first tier of analysis, four variants within the following genes: *ATP8B2*, *MAP3K3*, *NIPBL*, and *MSL3*, were prioritized for further investigation. The population frequency for all selected variants was zero in gnomAD (database, v.3) [3] and in in-house datasets of >3500 WES of Polish individuals. Selected variants were validated in the proband and studied in all available family members (i.e., both parents and maternal grandparents) by amplicon deep sequencing performed using the Nextera XT Kit (Illumina) and sequenced on a HiSeq 1500 (Illumina). All selected variants were revealed to be inherited, thus disqualified as causative of the patient’s phenotype.

Second trier analysis exposed a hemizygous deletion in the Xq24 region that included two genes: *UBE2A* (exon 1–3) and *CXorf56* (exon 1–6) (Figure 2). A multiplex allele specific PCR test was designed to validate the presence of the detected deletion (two pairs of primers were used: for amplification, a control one, which amplified the sixth exon of the *UBE2A* gene, and a second, which was settled within the deletion region, i.e., the second exon of the *UBE2A* gene). With this method, we confirmed the presence of the deletion in the proband, but it was impossible to establish the mother’s carrier status.

The array-CGH test performed in a commercial laboratory on the proband’s DNA in 2015 did not reveal any genomic imbalances. After the reanalysis of the WES results, the array-CGH assay was again performed on the mother’s and her affected son’s DNA (Agilent SurePrint G3 CGH ISCA v2, 8x60K). An unbalanced profile with an interstitial loss of an approximately 35kb fragment of the long arm of chromosome X (part of the cytoband q24) was revealed in the patient and his mother. The deletion (arr[GRCh37] Xq24(118679488_118714408)x1) included a part of the *UBE2A* gene (exons 1 to 3, NM_003336) and the *CXorf66* gene. Maternal origin of the causative deletion in the proband was confirmed.

The parents signed a written informed consent form for the genotyping and consented to the publishing of all the data generated. The study received the approval of the Bioethics Committee of Wroclaw Medical University (code: KB-430/2018; date of approval: 23 July 2018).

## 3. Discussion

Contiguous genes syndromes are conditions caused by the deletion or duplication of multiple genes’ loci that are adjacent to one another. They are usually sporadic and may be detected by molecular analyses [4]. Clinical manifestation usually depends on the size of the chromosomal imbalance. Contiguous genes syndromes may comprise genes located on autosomal (i.e., 22q11.2 microdeletion syndrome and WAGR syndrome) or sex chromosomes (Xp21 deletion comprising genes responsible for the enzyme glycerol kinase deficiency, Duchenne muscular dystrophy, congenital adrenal hypoplasia, and intellectual disability) [4,5,6,7].

In the literature, only nineteen patients with a deletion of the *UBE2A* gene and Nascimento-type syndromic intellectual disability inherited in a recessive X-linked manner (MRXSN, OMIM: 300860) have been described. The *CXorf56* gene deletion causes a separate entity associated with X-linked intellectual disability (MRX107, OMIM: 301012). The characteristic features of a syndrome caused by intergenic deletion of both the *UBE2A* and *CXorf56* genes include moderate to severe intellectual disability, heart defects, urogenital system anomalies, and dysmorphic features [8,9,10]. This clinical picture corresponds to the signs observed in the presented patient. However, it appears plausible that this syndrome could be correlated with the loss of function of *UBE2A* only.

Urogenital anomalies such as hypospadias, cryptorchidism, and small penis connected to *UBE2A* gene defects have been observed [8,11,12]. Nicole de Leeuw described five patients with Xq24 deletion and VSD. The patient described here presented with similar urogenital findings and a heart defect (AVS and VSD). Hypotonia and severe psychomotor delay have also been previously described as features of *UBE2A* deficiency syndrome [11,12] and are a part of the phenotype of our patient. Contrast MRI of the brain was performed in the presented patient. Non-specific dysmyelination changes with vascular component and slight lateral ventricular dilatation were found. The literature on intergenic deletion in the Xq24 chromosome region includes information on white matter abnormalities on MRI [8,12,13,14].

Precise and early diagnosis of congenital anomalies and developmental delay is extremely important in order to provide adequate care for the patient and his family in terms of the risk of relapse. Currently, for patients with unexplained multiple congenital anomalies, the international consensus proposes chromosomal microarray as a first-line test [15]. Microarray testing for CNVs (copy number variations) is recommended as the initial evaluation for patients with multiple defects not specific to a well-delineated genetic syndrome. Clinical microarrays supersede the limitations of NGS algorithms in the detection of microcopy number variants especially on the X chromosome. Various next-generation sequencing-based tailored gene panels constitute the second line of testing. NGS has changed the approach to rare dysmorphic and multi-defects syndromes [16,17]. Whole exome sequencing (WES) will be considered if these first and second line tests cannot determine a definitive diagnosis. Such an approach enables the diagnoses of genetic etiology in 60% of children with moderate-severe intellectual disability accompanied by malformations and/or dysmorphic features [6,7]. Additionally, NGS is a potent tool to fully characterize the breakpoints of all types of balanced chromosomal rearrangements confirming gene disruption, which could account for the patient’s phenotype [18].

NGS technology is currently leading to the possibility of the identification of genetic syndromes by a process called reverse dysmorphology, i.e., the delineation of new syndromes primarily by genotype followed by the description of the phenotype [15,19]. Such an approach can be a very useful element in the diagnostics of genetic syndromes. The new approach, “genotype first”, points to the phenotype shared by all patients with the same variant/genotype. Reverse dysmorphology as a diagnostic process was described in a large group of patients when high-resolution CGH studies were employed in diagnostic testing, hence allowing linking new critical chromosomal regions to new phenotypes and dysmorphic syndromes [15,19]. Consistently, through NGS, a pathogenic variant in a gene known to cause disease might be identified, prompting clinicians to re-evaluate the phenotype and make the correct diagnosis, compatible with reverse phenotyping.

WES produces vast amounts of data; however, establishing a causative relationship between a genomic variation and a particular disorder still remains a challenge. There have already been several studies demonstrating that reanalysis of exome data increases the diagnostic yield by about 10%. Generally, the reasons include constantly evolving knowledge about gene-disease associations, improvement of bioinformatic tools, growing expertise in medically interpreting genomic variation, and better collaboration between international case sharing databases and improved communication between clinical geneticists and their laboratory colleagues.

Since a first-line CGH analysis performed in a commercial facility did not reveal any genomic imbalances, WES data were screened at first for point, potentially pathogenic, variants. However, all indicated variants were revealed in the WES study to be inherited and disqualified as causative, so we decide to perform thorough reanalysis without any primary hypothesis (i.e., lack of CNVs), which resulted in the identification of the causative hemizygous intergenic deletion. This finding highlights the value of reanalysis, particularly a hypothesis-free one. We also underscore the importance of frequently redesigning and updating the coverage of clinical microarrays. The identification of the cause of the disease was additionally valuable, because of the possibility to accurately estimate the risk of subsequent offspring developing the disease (the mother is an asymptomatic carrier).

In conclusion, we present a case of a patient with intellectual disability, dysmorphic features, and congenital anomalies of the heart and urogenital system caused by a deletion in the Xq24 chromosome region that includes the *UBE2A* and *CXorf56* genes. 

Our case demonstrates the importance of making a diagnosis by pinpointing the underlying genetic defect, not only for the sake of the patient, but also his entire family. Establishing the proband carrier status of the mother permitted adequate assessment of the recurrence risk of the syndrome in her offspring. Periodic reanalysis of the exome data in the context of an individual’s phenotype, despite “exhausting” existing methods of genetic testing, should become a standard for patients with unexplained intellectual disability, dysmorphic features, and/or multiple congenital anomalies.

The combination of new genomic testing tools and techniques including array-CGH and NGS are making dysmorphology a very exciting and dynamic discipline of clinical genetics. The parallel improvement in both phenotyping and genotyping and their reciprocal interaction can facilitate making molecular diagnosis in dysmorphology and improve our knowledge on the pathogenesis of a number of diseases. 

Moreover, our report expands our knowledge about the genotype and clinical phenotype of this syndrome.

## Figures and Tables

**Figure 1 genes-12-00350-f001:**
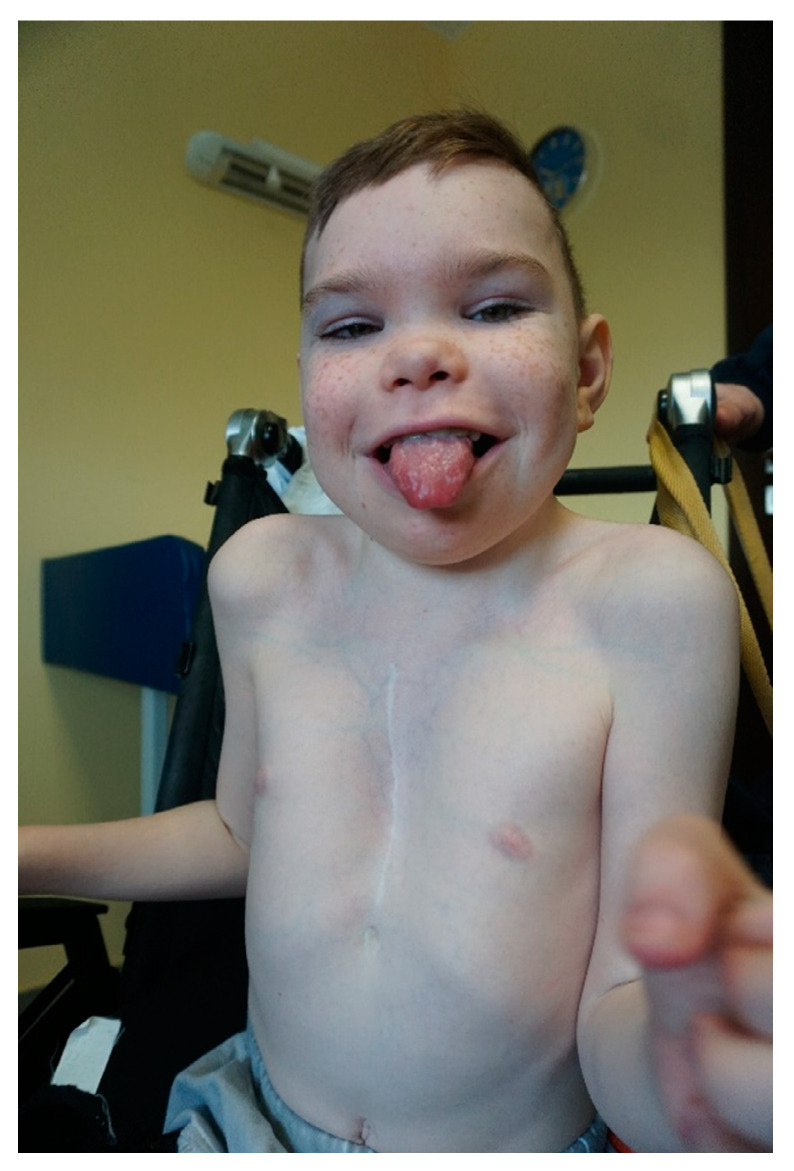
Features observed in the investigated subject.

**Figure 2 genes-12-00350-f002:**
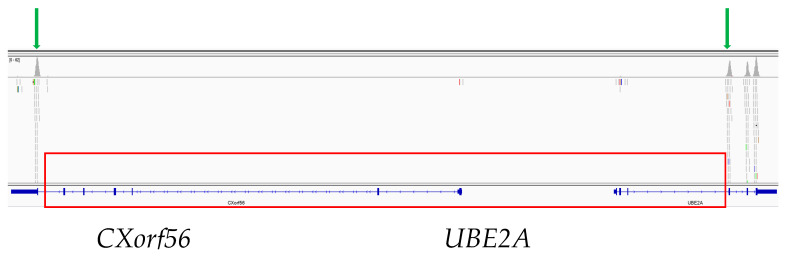
Graphical presentation of the partial deletion of the *UBE2A* and *CXorf56* genes created in Integrative Genomics Viewer v.2.8. The red frame denotes the deleted region. The green arrows denote the first remaining exons of the *CXorf56* and *UBE2A* genes, respectively.

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
