# Peer review of "The Role of the Reanalysis of Genetic Test Results in the Diagnosis of Dysmorphic Syndrome Caused by Inherited Xq24 Deletion including the UBE2A and CXorf56 Genes"

_genes, 2021, doi:10.3390/genes12030350_

Round 1

Reviewer 1 Report

The report presents a patient with specific dysmorphic, neurodevelopmental and neurological features whose first tier genomic analyses failed to identify a specific genotype. This report shows how re-analyzing WES data can provide more insights in a later date as the authors re-analyzed the sequencing data and this time were able to identify the causative genomic rearrangement (Xq24 deletion).

The workflow followed for the experimental design is clear and easy to understand.

They also present novel MRI findings for this deletion.

There are some concerns to highlight:

Major concerns:

  • The authors state: "Bioinformatic analysis of raw WES data and variants prioritization were performed as previously described [1]. Reads were aligned to the hg38 67 reference genome sequence"

Which reference genome was used in the previous analysis? This would be important in case it was not Hg38, to point others to use more the latest genome assembly and ditching Hg19 which is already outdated, specially for regions prone to genomic rearrangements. It is critical to add more details on the previous and new analyses as the readership need to know how re-analyses of previous WES could provide a better diagnosis, and in addition, this is one of the most interesting aspects of the article.

  • More details about the ethics approval and consent are required.

Reviewer 2 Report

The authors describe results of reanalysis of genetic tests in a patient with ID and dysmprphic features. This is an important message, because frequent reanalysis of genetic tests will provide more and more etiological diagnosis in patients with ID and/or dysmorphisms and/or congenital anomalies.

I think the English  language can be improved.

title: delete "the role of" : Reanalysis of genetic tests results in.......

line 38: agressive behaviour(s)

line 44: patients-> the patient

line 45: prominent supraorbital ridges narrow?????? unclear

line 79: within the deletion region

line 90; denotes the first

line 128-130: this sentence does not go well

line 13860% of children...

line 144: syndromes

line 146: be a very....

line 151: variant(s)

line 155: between a genomic..

line 165-166: this sentence does not go well

line 167-169: this sentence does not go well

line 175: recurrence...

line 185: ..genetic=genotype...

Reviewer 3 Report

Summary Statement:

Ewelina et al. describe an affected male case presenting with a neurological and dysmorphic phenotype. The described proband harbors a hemizygous maternally inherited Xq24 deletion that is ~35-kb in size (approximated by HD-aCGH) and extends over chrX:118,679,488-118,714,408 (hg38) encompassing parts of two previously disease-implicated genes, UBE2A and Cxorf56. Initial analysis of diagnostic aCGH and Exome sequencing did not indicate the presence of the Xq24 deletion which was later identified upon reanalysis of the Exome sequencing data. The proband presents with clinical features concordant with UBE2A deficiency syndrome (or Syndromic X-linked Mental retardation, Nascimento-type) [MIM# 300860]. UBE2A deficiency syndrome has been previously associated with UBE2A loss of function variants such as hemizygous Xq24 micro-deletions, and nonsense variants in the gene.

Overall:

The authors do a good job in describing the case clinically and providing clear background information on the implicated locus and phenotype and we think that this is worthy of publication in your journal. The association of the Xq24 deletion and the observed phenotype in the proband is strongly backed up with evidence from the literature. This Case Report confirms previous associations of the UBE2A locus with a neurological and dysmorphic disease phenotype and provides further insight into the clinical course of this disease in older individuals.

Major comments:

1) The title: "The role of reanalysis of genetic tests results in diagnosis of 3 dysmorphic syndrome with intellectual disability. Additional 4 case of contiguous genes deletion in Xq24 chromosome region 5 including UBE2A and CXorf56 genes" is too long and should be shortened.

2) This manuscript needs to be proofed as there are still typos. Examples: Page 2, line 46 there is a comma instead of a period.  There are also extra spaces at multiple locations, see for instance, Page 1, Line 23.

3) A full list (with citations) of previously published Xq24 deletions would be beneficial for a reader trying to learn about the science. As an example the paper by Honda et al. (Honda, S., Orii, K., Kobayashi, J. et al. Novel deletion at Xq24 including the UBE2A gene in a patient with X-linked mental retardation. J Hum Genet 55, 244–247 (2010). https://doi.org/10.1038/jhg.2010.14) is not mentioned.

Minor comments:

Pg.1 Line 20: Genes should be italicized as per international gene nomenclature.  

Pg.3, L102: The authors refer to the phenotype as a contiguous gene syndrome, which is of concern because the identified deletion only partially encompasses the implicated genes. Additionally, given the strong evidence available for the involvement of one of the two involved genes, UBE2A, but not as much for the other, cXorf56, it appears plausible that this syndrome is likely correlated to the loss of function of UBE2A only. Therefore, authors should be careful with the designation of a “contiguous gene syndrome” in this case report without further evidence for the involvement of cXorf56 is identified.

Pg. 4 L125-126: The authors claim that no detailed MRI information has been reported in previous cases of intergenic Xq24 deletions. However, a number of publications including Honda et al. (2010), Thunstrom et al., de Leeuw et al. and Czeschik et al. discuss some of their intergenic Xq24 deletion cases MRI findings at reasonable depth.

Pg. 4 L128-142: The authors highlight the importance of NGS data reanalysis to solve unsolved cases which is essential. However, they do not equally underscore the importance of frequently redesigning and updating the coverage of clinical microarrays. Clinical microarrays supersede the limitations of NGS algorithms in the detection of micro copy number variants especially on the X chromosome.

Pg. 4, L164: “pinpointed” à replace with ACMG-approved terminology.
